# Assessment of radio-activation using spectroscopy in medical linear accelerators

Na Hye Kwon[1], Young Jae Jang[2,3], Suah Yu[3,4], Hanjin Lee[5], Dong Hyeok Choi[1], So Hyun Ahn[6], Kum Bae Kim[3], Jin Sung Kim[1,7], Dong Wook Kim[1]*, Sang Hyoun Choi[3]*

1 Department of Radiation Oncology, Yonsei Cancer Center, Heavy Ion Therapy Research Institute, Yonsei University College of Medicine, Seoul, Republic of Korea, 2 Department of Accelerator Science, Korea University, Sejong, Republic of Korea, 3 Research Team of Radiological Physics and Engineering, Korea Institute of Radiological and Medical Sciences, Seoul, Republic of Korea, 4 Department of Radiological Science, Kangwon National University, Samcheock, Republic of Korea, 5 Department of Radiological Emergency Preparedness, Korea Institute of Nuclear Safety, Daejeon, Republic of Korea, 6 Ewha Medical Research Institute, School of Medicine, Ewha Womans University, Seoul, Republic of Korea, 7 Oncosoft Inc., Seoul, Republic of Korea

☯ These authors contributed equally to this work.
* joocheck@gmail.com (DWK); shchoi@kirams.re.kr (SHC)

**Data Availability Statement:** All relevant data are within the manuscript and its Supporting Information files.

**Funding:** This research was supported by the National Research Council of Science and

## Abstract

In radiotherapy, when photon energy exceeding 8 MV is utilized, photoneutrons can activate the components within the gantry of the linear accelerator (linac). At the end of the linac's lifecycle, radiation workers are tasked with its dismantling and disposal, potentially exposing them to unintentional radiation. This study aims to identify and measure the radioisotopes generated by this activation through spectroscopy, and to evaluate the effective dose rate. We selected nine medical linacs, considering various factors such as manufacturer (Siemens, Varian, and Elekta), model, energy, period of operation, and workload. We identified the radionuclides in the linac head by employing an in situ high-purity germanium (HPGe) detector. Spectroscopy and dose-rate measurements were conducted post-shutdown. We also measured the dose rates at the beam-exit window following irradiation with 10 MV and 15 MV photon beams. As a result of the spectroscopy, we identified approximately 20 nuclides including those with half-lives of 100 days or longer, such as $^{54}$Mn, $^{60}$Co, $^{65}$Zn, $^{122}$Sb, and $^{198}$Au. The dose rate measurements after 10 MV irradiation decreased to the background level in 10 min. By contrast, on 15 MV irradiation, the dose rate was 628 nSv/h after 10 min and decreased to 268 nSv/h after 1.5 hours. It was confirmed that the difference in the level of radiation and the type of nuclide depends on the period of use, energy, and workload. However, the type of nuclide does not differ significantly between the linacs. It is necessary to propose appropriate guidelines for the safety of workers, and disposal/move-install should be planned while taking into consideration the equipment's energy usage rate.

## Introduction

A linear accelerator (linac) is a representative treatment equipment used in radiotherapy, including tomotherapy, Cyberknife, and Gamma Knife. According to data from the Radiation Safety

Technology (NST) grant by the Korea government (MSIT) (No. CAP22041-000) & Korea Institute for Advancement of Technology(KIAT) grant funded by the Ministry of Trade, Industry and Energy (MOTIE) (P20026103) & Korea Institute of Radiological and Medical Sciences (KIRAMS) funded by Ministry of Science and ICT (MSIT) (No. 550572-2024) & Korea Foundation of Nuclear Safety (KoFONS) granted by the Nuclear Safety and Security Commission (NSSC) of the Republic of Korea (No. 2205013).

**Competing interests:** The authors have declared that no competing interests exist.

Information System, as of 2023, 103 medical institutions in South Korea use radiation generators. [1]. The linac is the most commonly installed and used among various types of radiotherapy equipment [2]. The number of linacs in South Korea with maximum MV energy are 64 for 6 MV, 104 for 10 MV, and 73 for 15 MV. When an energy of 8 MV or more is used, various parts located in the head of the linac are continuously irradiated, resulting in a phenomenon known as photon-induced activity. This activation does not occur at energies below 8 MV.

When photon energy is greater than the binding energy of the nucleus, a photonuclear reaction occurs, and neutrons are emitted. The emitted neutrons are recaptured via a reaction that emits gamma rays. The corresponding linac equipment becomes radioactive as these processes are repeated for 10–15 years [3–5].

The radiation levels of linacs vary depending on various conditions, such as the energy used, number of patients, and treatment technique. Linacs are typically discarded after 10–15 years of operation. Although different institutions follow different protocols regarding equipment disposal, most of them conduct radiation evaluations after removing the equipment [6]. However, the activation effect must be considered even when transferring a linac used in one institution to another. Radiation workers are inevitably exposed to radiation when removing equipment from its treatment room after the beam is shut down.

Guidelines are needed to minimize the effect of radiation exposure of a linac based on the dose rate assessment and information on the gamma-ray-emitting nuclides generated by the linac itself. Regulations on radioactive waste components were established in Korea in 2022, and self-disposal is permitted if the radioactive waste is less than 0.1 uSv/h at a distance of 1 cm (after subtracting the background average value). However, there are no regulations yet for the case of moving the equipment itself, and exposure in the process is not considered.

In previous studies [3,4,7,8], the nuclides generated in a linac by the activation effect were investigated. They studied models that had been in use for more than ten years. In addition, the nuclides generated after irradiating high-energy photon beams of 15 or 18 MV were experimentally investigated. However, only a few studies have considered characteristics such as the workload of the actual equipment, period of use, and number of patients as possible factors. Studies on radiation occurring in various parts of the linac head are available. Lee et al. and Juste B et al. [9–12] conducted a study using the Monte Carlo method by considering only the main components, such as targets and filters. The effect of the radiation in the internal parts of the linac head on workers when dismantling or relocating equipment for disposal have not been considered.

In this study, our objective was to assess the impact of activation in linacs through the analysis of radionuclides and measurement of gamma dose rates. We evaluated the linacs used in five medical institutions and studied the difference in the level of radiation depending on actual device use history and conditions. A portable high-purity germanium (HPGe) detector and a survey meter were used to identify gamma-emitting nuclides and evaluate radioactivity without additional disassembly. Accordingly, guidelines for the removal and disposal of linacs were established.

## Materials and methods

### Information of medical linacs and facilities

We measured and evaluated the linac activation in collaboration with five medical institutions. We selected nine linacs considering different manufacturers (Elekta, Siemens, and Varian) and the energy used (6, 10, and 15 MV). The activation phenomenon occurs when the photon energy for treatment exceeds 8 MV. Therefore, we selected a linac with a minimum energy of 10 MV. We summarized the information such as the manufacturer, model name, number of

**Table 1. Information of linacs and facilities.**

| Manufacturer | Elekta | | | Siemens | | | Varian | | |
|---|---|---|---|---|---|---|---|---|---|
| Linear Accelerator No. | 1 | 2 | 3 | 4 | 5 | 6 | 7 | 8 | 9 |
| Model | Infinity | Versa HD | Versa HD | Oncor Impre -ssion Plus | Oncor Expre -ssion | Oncor Impre -ssion | Clinac iX | Clinac iX | Trilogy |
| Number of patients [1/day] | 30 | 110 [1/month] | 100 [1/month] | 20–30 | 50 | 10 | 45 | 40 | 60 |
| Workload [Gy/ week] | 300 | 700 | 730 | 300 | 400 | 100 | 500 | 500 | 712 |
| Period of operation | 6 mos. | 8 yrs. | 3 yrs. | 15 yrs. | 15 yrs. | 15 yrs. | 8 yrs. | 10 yrs. | 11 yrs. |
| Energy [MV] | 6, 10 | 6, 10, 15 | 6, 10, 15 | 4, 10 | 6, 15 | 6, 10 | 6, 10 | 6, 10 | 6, 15 |
| 4, 6 MV | 80% | 91% | 82% | 70% | 50% | 100% | 80% | 80% | 80% |
| 10 MV | 20% | 9% | 18% | 30% | - | 0% | 20% | 20% | - |
| 15 MV | - | 0% | 0% | - | 50% | - | - | - | 20% |
| HPGe detector used for spectroscopy | Falcon 5000 | Trans-SPEC–DX100 | | Falcon 5000 | Trans-SPEC–DX100 | | Falcon 5000 | | |

treated patients, workload (per week or month), operation period, and energy usage ratio obtained according to the institution and the equipment in Table 1. The period of use is defined from when the equipment is installed to when the measurement is performed. Linacs (1), (4), (7), and (8) as mentioned in Table 1 are installed in the same institution with operation periods between 6 months and 15 years. All four devices mainly operate with 6 MV of energy. Linacs (2) and (3) manufactured by Elekta can use 6, 10, and 15 MV of energy. However, 6 MV and 10 MV are used for actual patient treatment. Even in the case of 10 MV, a radiation phenomenon is not expected to occur because a flattening filter-free beam is used. Linac (5), an Oncor Expression equipment manufactured by Siemens, uses 15 MV of energy at a high rate of approximately 50% and has an operation period of approximately 15 years. For linac (6), an Oncor Impression equipment, 6 MV and 10 MV of energy are available. However, since 2020, only 6 MV of energy has been used for patient treatment. For equipment quality analysis, from 2020 to March 2022, 10 MV was used, and after April 2022, only energy of 6 MV was used. Linac (9), a Trilogy model manufactured by Varian, treats an average of 60 patients per day and uses 15 MV of energy at a rate of 20%. Linacs (5) and (9) were disposed after measurement.

## Spectroscopic measurements using portable HPGe detector

We evaluated the radiation of the linacs using a portable HPGe detector and identified the generated nuclides. As shown in Fig 1, an HPGe detector was placed in front of the beam window exit for the spectroscopic measurements, with both the jaw and multi-leaf collimator open. We rotated the gantry 90° and adjusted the height of the sofa so that the center of the detector was in the center of the beam exit window. Spectra were acquired for 1 hour immediately after the patient had finished treatment. The time required to set up the detector was less than 10 minutes. However, for the Siemens linac (5), measurements were performed after 5 hours of treatment according to the guidance of the hospital administrator.

The spectra were measured using Falcon 5000 (Canberra Inc., USA) and Trans-SPEC–DX100 (Ametek Ortec, USA) detectors. We utilized both Canberra and Ortec systems for

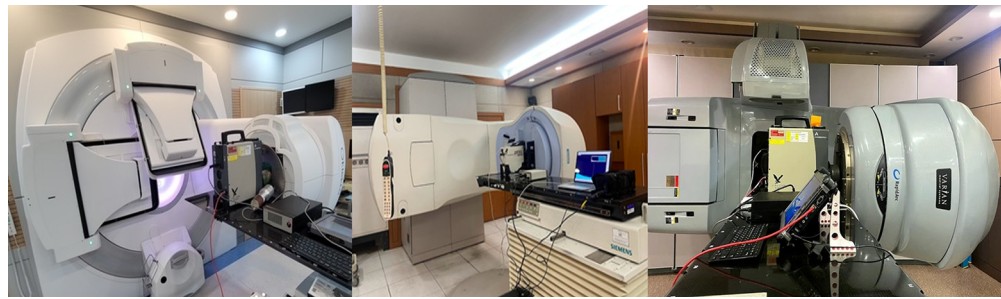

**Fig 1. Set-up of HPGe detector for spectroscopic measurements.**

measurements, each employed for spectroscopy of a different linac of the facility. Table 2 lists the models, detection efficiency, energy resolution, and analysis software used for each of the two detectors. For energy & efficiency calibration, we utilized values based on the certified reference material of $^{152}$Eu, keeping in mind the potential inaccuracies due to the variance in density, shape, and size of the iron-based radioactive waste generated from the linac. Prior to spectra acquisition, we used a $^{60}$Co authoritative source for in-situ re-checking of the energy-channel correlation, leveraging the identified gamma peaks at 1.17 and 1.3 MeV. The acquired spectra were analyzed to determine the gamma-emitting nuclides generated by the equipment.

## Dose rate measurements using survey meter

The dose rate was measured at the beam exit window using a survey meter (Thermo FH 40 G-L10 Survey Meter; Thermo Inc., Germany). This gamma meter has an internal proportional counter tube. We used a dose-rate calibration factor of 1.1, as provided by the Korea Research Institute of Standards and Science (KRISS), the National Metrology Institute. The measurement of the dose rate with the survey meter was conducted at a distance of 0 cm from the front of the beam exit window (Fig 2). First, we measured the dose rate after the end of the treatment to consider the activation level depending on the equipment condition according to usage history. In addition, we intentionally irradiated 500 monitor units (MU) with photon beams of 10 MV and 15 MV energy and measured the change in the dose rate over time. The background radiation dose rate was measured inside the treatment room before beam irradiation.

## Results

### Results of spectroscopic measurements: Linac equipment

We confirmed that in the case of equipment using a photon beam of 10 MV or more, as a result of spectroscopy measurement, nuclides detected ranged from 8 to 18. The spectra were acquired using the HPGe detector, and the nuclide information was confirmed based on the

**Table 2. Specifications of portable HPGe detector for gamma-ray spectroscopic measurements.**

| Manufacturer/Model | Canberra/Falcon 5000 | Ortec/Trans-SPEC–DX100 |
|---|---|---|
| **Efficiency** | **18%** | **40%** |
| Gamma energy resolution (Full Width Half Maximum, keV) | | |
| 122 keV | 1.0 keV (0.82%) | 1.6 keV (1.31%) |
| 1332 keV | 2.0 keV (0.15%) | 2.3 keV (0.17%) |
| Software | Genie2000, ISOCS | Gamma Vision |

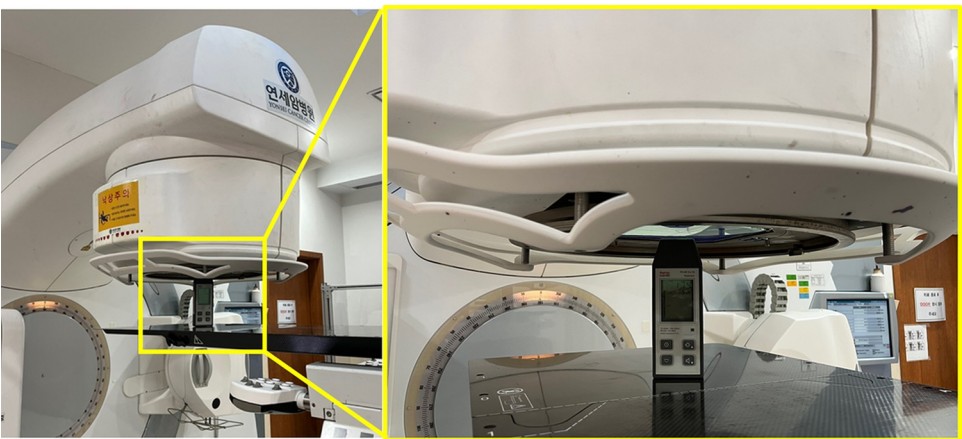

**Fig 2. Set-up of HPGe detector for spectroscopic measurements.**

counts per second (cps) according to the energy. As shown in Fig 3, the main peak and name of the nuclide are shown in **Bold,** while the curve after the secondary peak is shown in gray color. The peak with green color is the natural isotope. Fig 3 shows the results for the Elekta (3), Siemens (5), and Varian (9) linacs. We selected those linacs using a high energy of more than 10 MV.

Table 3 summarizes the results of the gamma-ray spectroscopy for the radiation evaluation of the linacs. We have organized the cps and isotopic information by energy into a table, which is included in S1 Table. In the acquired spectra, except for natural radionuclides such as $^{40}$K, we marked the artificial radionuclides generated by radioactivity with 'D' (Detected). We observed a relatively large number of nuclides for the equipment using 10 and 15 MV, with a larger portion having half-lives of more than 100 days. As observed from the result of the spectroscopic analysis of the linac manufactured by Elekta, the number of nuclides detected in the linac (1) was the lowest. For the linacs (2) and (3), the types of nuclides detected were the same, except for $^{59}$Fe and $^{99}$Mo. The cps level of the spectrum of linac (3) (approximately 100 cps) was higher than that of (2), even though the tests were performed under same conditions.

In the case of linacs (4), (5), and (6) from Siemens, the history of use for each piece of equipment was the most diverse compared with any other linac of the manufacturer. The resulting influence can be confirmed in the measurement results. Many of the nuclides identified in linac (4) were similar to those in (5), but the cps level was 1/10 times lower at the peak of $^{60}$Co. In the case of linac (5), with the highest 15 MV usage rate, the most radionuclides were detected among the nine of equipment. The case of linac (6) using only 6 MV, it can be judged that it rarely activated and that the maximum cps was 410, indicating a shallow level of radioactivity and the lowest number of detected nuclides.

Compared to detection results for linac (8), linac (7) that treated more patients and used 10 MV at a higher rate detected relatively more nuclides. Linac (9) with a high usage rate of 15 MV showed long-lived nuclides with half-lives of more than 100 days, including $^{54}$Mn, $^{57}$Co, $^{60}$Co, and $^{65}$Zn.

## Gamma dose rate measured in front of the beam exit

Before the spectroscopy, we measured the gamma dose rate in front of the beam exit to evaluate the radiation levels indirectly. Except for that of linacs (5) and (9), dose-rate measurement data were under 0.2 uSv/h. The gamma dose rates measured after patient treatment were 0.532

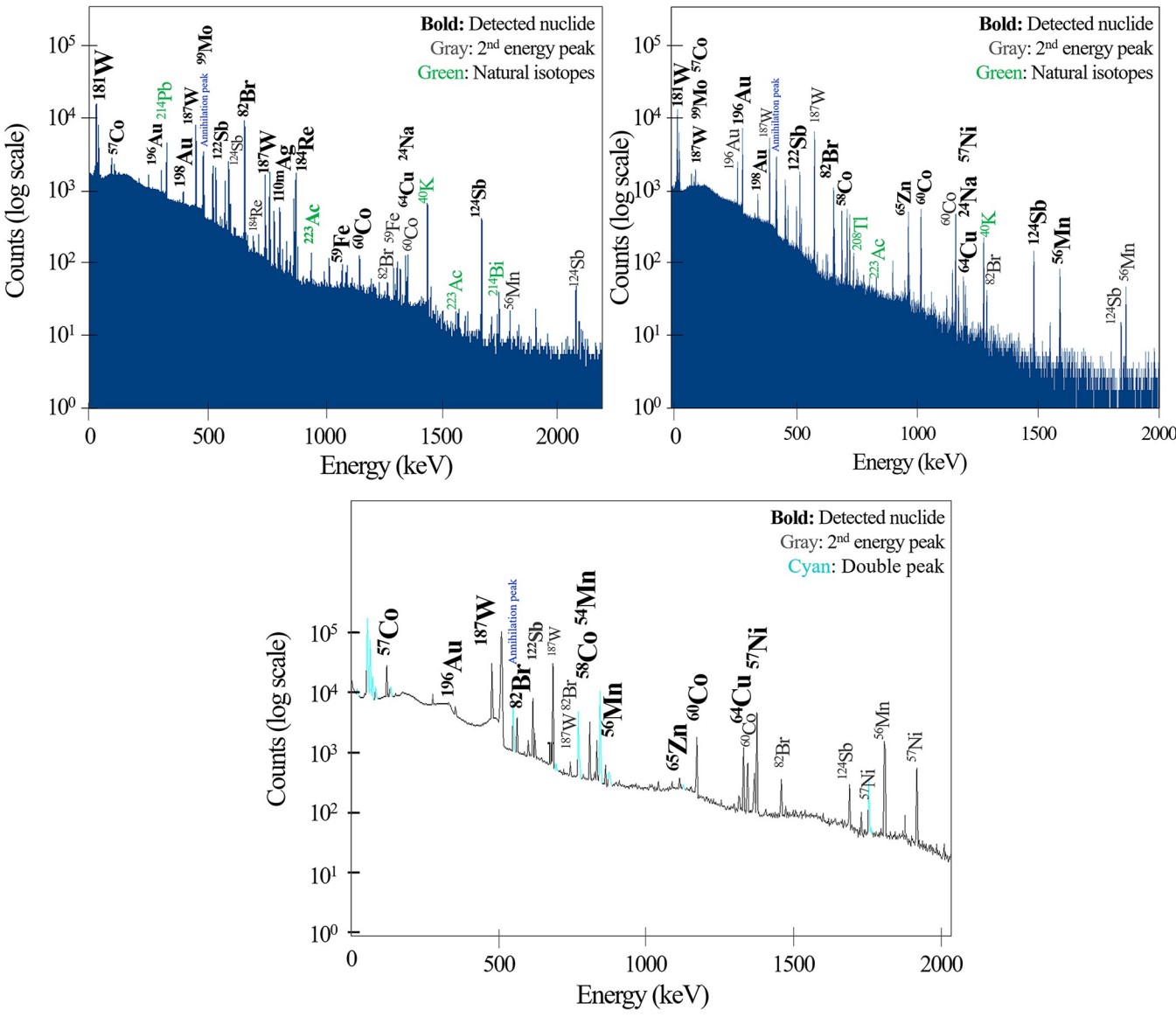

**Fig 3.** Acquired spectra of linacs: (left) Elekta linac (3), (center) Siemens linac (5), (right) Varian linac (9).

uSv/h for linac (5) and 2.773 uSv/h for linac (9). For linac (5), we measured the gamma dose rate after approximately 5 hours. The gamma dose rates were measured after irradiating the 500 MU with a photon beam (10 MV and 15 MV, respectively). The gamma dose rate decreased logarithmically over time. As shown in Fig 4, immediately after 500 MU irradiation using a 10 MV photon beam, the 0.702 uSv/h gamma dose rate decreased to the background level of 0.115 uSv/h within 10 minutes. The gamma dose rate decreased logarithmically, and $R^2$ was 0.9731. Considering the irradiation of 15 MV, the gamma dose rate decreased from 1.2 uSv/h to 0.268 uSv/h even after 1.5 h flows, and $R^2$ was 0.9846.

## Discussion

In the spectroscopy results of the linacs, variations were observed in the measurements, even amongst equipment from the same manufacturer or under similar conditions. Specifically,

**Table 3. Radionuclides data from spectroscopy results and comparison with previous studies.**

| Manufacturer | | | Elekta | | | | Siemens | | | | Varian | | | |
|---|---|---|---|---|---|---|---|---|---|---|---|---|---|---|
| Linac No. / Reference paper No. | | | Our data | | | Ref. paper | Our data | | | Ref. paper | Our data | | | Ref. paper |
| | | | 1 | 2 | 3 | Fisher HW [4] | 4 | 5 | 6 | Fisher HW [4] | 7 | 8 | 9 | Fisher HW [3,4] Fujibuchi T [7] |
| Nuclide | Key energy (keV) | Half-life | Radionuclides results | | | | | | | | | | | |
| $^{24}$Na | 1368.67 | 15.02 h | | D | D | D | D | D | | D | D | D | D | D |
| $^{28}$Al | 1778.98 | 2.240 m | | | | D | | | | D | | | | D |
| $^{51}$Cr | 320.08 | 27.70 d | | | | D | | | | | | | | |
| $^{54}$Mn | 834.85 | 312.2 d | | D | D | D | | D | | D | | | D | D |
| $^{56}$Mn | 846.77 | 2.579 h | D | D | D | D | D | D | D | D | | D | D | D |
| $^{57}$Co | 122.06 | 270.9 d | | D | D | D | | D | | D | | D | D | D |
| $^{57}$Ni | 1377.64 | 35.65 h | | D | D | D | | D | | D | | D | D | D |
| $^{58}$Co | 810.76 | 70.80 d | D | D | D | D | | D | | D | | D | D | D |
| $^{59}$Fe | 1099.22 | 44.63 d | | D | D | D | | D | | D | | | | D |
| $^{60}$Co | 1173.23 1332.48 | 5.271 y | | D | D | D | D | D | D | D | | D | D | D |
| $^{62}$Cu | 511 1173.02 | 9.740 m | | | | D | | | | D | | | | D |
| $^{64}$Cu | 1345.77 | 12.70 h | | D | D | D | | D | | D | | | D | D |
| $^{65}$Zn | 1115.52 | 243.7 d | | | | D | D | D | | D | | | D | D |
| $^{82}$Br | 776.51 | 35.30 h | | D | D | D | D | D | | D | D | D | D | D |
| $^{99}$Mo | 140.50 | 66.02 h | | D | D | D | D | D | | D | | | | |
| $^{116m}$In | 1293.5 | 54.30 m | | | | | | | | | | | | D |
| $^{122}$Sb | 563.93 | 2.700 d | | D | D | D | D | D | | D | D | D | D | D |
| $^{124}$Sb | 602.72 | 60.21 d | D | D | D | D | D | D | | D | D | D | D | D |
| $^{181}$W | 57.53 | 121.2 d | | D | D | | | D | | | D | D | D | D |
| $^{184}$Re | 903.27 | 38.00 d | D | D | D | D | | | | | | | | |
| $^{187}$W | 685.81 | 23.83 h | D | D | D | D | D | D | | D | D | D | D | D |
| $^{196}$Au | 355.73 | 6.183 d | D | D | D | D | D | D | | D | | | D | D |
| $^{198}$Au | 411.80 | 64.80 d | | D | D | | | D | D | | | | D | D |
| $^{203}$Pb | 279.19 | 52.93 h | | D | D | | | | | | | | D | |

Elekta's linac (1), which was installed and operated for six months, detected fewer nuclides compared to linacs (2) and (3). From the perspective of workload, linac (3), which had a higher workload, detected more radionuclides than linac (1). These results are in accordance with the

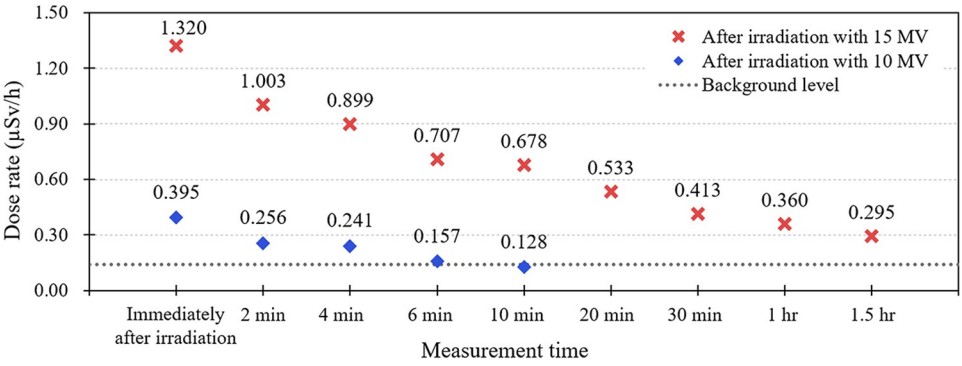

**Fig 4. Results of dose rate measurements in front of beam exit window.**

study by YJ Jang et al. [13]. When comparing linac (2) and (3), there was no significant difference in the types and numbers of detected nuclides, but differences in cps were confirmed according to the energy ratio used. The cps level of linac (3), for which the 10 MV energy ratio was 18%, was relatively higher than that of linac (2).

As the linac (4) and (5) evaluations were under similar conditions, such as the workload and the operating period, the number of nuclides detected in linac (5) using 15 MV of energy was larger. Only two nuclides were observed in Siemens linac (6) ($^{56}$Mn, $^{198}$Au). Because the linac uses only 6 MV of energy, we did not expect radioactivity. The maximum energy used in the treatment and the ratio of each energy were confirmed to affect the activation level significantly. Moreover, even if the same energy was used, the relative cps level increased if the ratio of 10 and 15 MV was high.

We compared the results of our study with those of previous research [3,4,7] in Table 3. Despite the generational evolution of the equipment, the nuclides detected across different linac models exhibited minimal disparities. Contrary to prior studies, our evaluation was based on actual treatment conditions and usage, without any intentional irradiation. Consequently, our results did not identify nuclides with extremely short half-lives, such as $^{62}$Cu (half-life 9.74 min) and $^{28}$Al (half-life 2.24 min). Both our measurements and the outcomes of preceding studies predominantly detected nuclides with half-lives under 100 days. Four nuclides with half-lives exceeding 100 days ($^{54}$Mn, $^{57}$Co, $^{60}$Co, $^{65}$Zn) were commonly identified, irrespective of the manufacturer. As for the nuclide $^{181}$W, it mainly possesses a low energy peak under 100 keV, which appears to be excluded in the study by Fisher et al.

Even in the case of high-energy (15 MV) utilization approaching 50%, most detected nuclides had half-lives shorter than several days, and fewer than six had half-lives greater than 100 days. Because the cps values of the mid-and long-lived nuclides (with half-lives longer than 60 days) were small, the degree of radioactivity was expected to be mainly affected by the short-lived nuclides.

We measured the dose rate in front of the beam exit window to evaluate the level of activation using a convenient way. As the components that were activated over a long period were shielded with the gantry head (lead or iron material), the measured data acquired at the beam exit window did not represent the radiation level of the parts inside the linac head. However, when a piece of equipment is previously installed or moved for disassembly/disposal, the workers did not disassemble the gantry head itself, and worker exposure is expected to be limited in front of the beam exit window. Therefore, we conservatively evaluated the degree of worker exposure based on the dose rate measurement values.

Based on the measured dose rate shown in Fig 4, we conservatively calculated the annual exposure dose of workers. We calculated the annual Dose equivalent (H) based on Eq (1), where D is the measured dose rate per hour, and T is the working time. The working time (T) was set considering the standard working hours of workers: 8 hours per day, 5 days per week, and 50 weeks per year. According to the measurement results for the 15 MV equipment, if work starts immediately after the patient's treatment is finished, the exposure is 2.4 mSv annually as following D is 1.2 uSv/h. However, if work starts after 1.5 h the exposure is 0.54 mSv per year. The two devices for which the dose rates were measured (Fig 4) were Elekta and Varian linacs, and both devices use only 10 MV for patient treatment.

$$H \ (\text{mSv/y}) = D \ (\text{uSv/u})*T \ (8 \ \text{hours}*5 \ \text{days}*50 \ \text{weeks}) \qquad \text{Eq(1)}$$

The rapid decrease in the dose rate was because the overall level of radiation was low, even though the measurement was performed after intentional irradiation. In addition, for equipment with high utilization rates of 15 and 10 MV, we consider that the measured dose rate will

be higher. Based on the dose rate for the linac equipment using 15 MV energy at 20% to 50% of linac (5) in the results section, the dose rate after treatment was observed to be as high as 2.773 uSv/h. For linac (9), even after approximately 5 h, it was as high as 0.532 uSv/h. When handling radioactive waste, the annual allowable dose for workers is 20 mSv. We believe exposure through a linac or its beam exit window is an insignificant risk factor. However, we expect that unnecessary exposure can be reduced given sufficient time, depending on the energy used before starting work after the Linac shutdown.

## Suggestion for linac move-install/disposal

In Korea, guidelines for disposing of linear accelerators in 2022 were recently enforced. However, the following clearance standards are limited to the individual evaluation of parts generated after dismantling the linear accelerator. (Acceptable level: average value + 0.1 uSv/h or less dose rate at a distance of 1 cm) When the linac is discarded or installed in another institution, the degree of activation of the linac and the amount of exposure to radiation workers must be considered. Gamma spectroscopy applied to linac measurements has a limitation: it can only assess nuclides. To calculate the radioactivity, corrections must be made for density, energy-specific measurement efficiency, and weight according to the standard material.

However, since most of the linac equipment are made of iron, there is a large difference from the CRM, which has a density of 1 and a weight level of grams. Thus, there may be an error in the measurement efficiency. Due to the high cost and complexity of analysis, acquiring gamma spectroscopy equipment for every hospital is challenging. Among the linac manufacturers, Varian provides work-related recommendations through their disposal and management guidelines [12]. Depending on the energy used Varian recommends waiting before starting disassembly/removal. For equipment using 10–14 MV of energy, work is recommended to commence at least 1 h after the end of the beam, and for equipment using photon energies over 15 MV, 12–24 h or one day after the end of the irradiation. Elekta recommends that scrapping for head parts should not exceed the maximum dose rate of 50 uSv/h [14]. They provide recommendations considering that the dismantling of a linac proceeds inside the treatment room, but they do not suggest any additional work start time for dismantling.

According to our measurement results and the manufacturer's recommendations, it is better to start work after roughly determining the level of radiation based on the measured value of the dose rate. In planning equipment disposal, it is important first to confirm the usage history of the institution, such as energy, number of patients, and operating load, and to determine the start time of work according to the ratio of energy use. If the 10 MV energy consumption rate is within 20%, it can be determined that the dose rate is less than 0.2 uSv/h after the equipment is turned off immediately. Even if it is higher than 20%, we can expect that it decreases to the background within a few hours. If 15 MV energy of linac was used, reducing the use rate before disposal seems to help lower the level of radiation, and it is judged that it will be 1/10 of the initial measurement value only after at least 3 hours.

## Conclusion

In conclusion, our study offers pivotal insights into the radio-activation of linacs operating at energies exceeding 10 MV, and provides practical guidelines for their safe disposal. Our findings underscore the significance of considering the energy usage, ratio, and operating load in evaluating radioactivity levels. Although the workload does influence dose rate and nuclide outcomes, we have found that the primary energy used and its ratio exert a more substantial impact on radio-activation. There were no discernible differences attributable to the equipment manufacturers. We recommend a waiting period following the cessation of the beam,

especially for equipment operating at 15 MV, and a conservative approach for linacs with energies below 10 MV. Our results contribute not only to the understanding of radio-activation but also lay the groundwork for establishing safety management standards for the disposal and relocation of therapeutic linacs.

## Supporting information

**S1 Table.**
(XLSX)

## Acknowledgments

We would like to express our heartfelt thanks to all those who have helped and supported with this article. Our deepest gratitude goes to Jihyun Yu and Byungchae Lee from the R&D Institute at Sae-An Enertech Corp., whose invaluable contributions to data analysis and insightful feedback have been indispensable.

## Author Contributions

**Conceptualization:** Na Hye Kwon, Kum Bae Kim, Jin Sung Kim, Dong Wook Kim, Sang Hyoun Choi.

**Data curation:** Na Hye Kwon, Young Jae Jang, Suah Yu.

**Formal analysis:** Na Hye Kwon, Young Jae Jang, Suah Yu.

**Funding acquisition:** Dong Wook Kim, Sang Hyoun Choi.

**Investigation:** Na Hye Kwon, Young Jae Jang, Dong Hyeok Choi.

**Methodology:** Na Hye Kwon, Young Jae Jang, Suah Yu, Hanjin Lee, So Hyun Ahn, Kum Bae Kim, Jin Sung Kim.

**Project administration:** Dong Wook Kim, Sang Hyoun Choi.

**Resources:** Hanjin Lee.

**Supervision:** Dong Wook Kim, Sang Hyoun Choi.

**Validation:** So Hyun Ahn, Jin Sung Kim, Dong Wook Kim, Sang Hyoun Choi.

**Visualization:** Na Hye Kwon.

**Writing – original draft:** Na Hye Kwon.

**Writing – review & editing:** Na Hye Kwon, Young Jae Jang, Hanjin Lee, Dong Hyeok Choi, So Hyun Ahn, Kum Bae Kim, Jin Sung Kim, Dong Wook Kim, Sang Hyoun Choi.

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
