## [Decision Letter · Decision Letter 0]

5 Nov 2023

PONE-D-23-29071Assessment of Radio-Activation Using Spectroscopy in Medical Linear Accelerators and Implications for Radiation Hazard Mitigation and Equipment DecommissioningPLOS ONE

Dear Dr. Choi,

Thank you for submitting your manuscript to PLOS ONE. After careful consideration, we feel that it has merit but does not fully meet PLOS ONE’s publication criteria as it currently stands. Therefore, we invite you to submit a revised version of the manuscript that addresses the points raised during the review process.

We look forward to receiving your revised manuscript.

Kind regards,

Mohamad Syazwan Mohd Sanusi

Academic Editor

PLOS ONE

Journal Requirements:

"This research was supported by the National Research Council of Science & Technology grant by the Korea government (Ministry of Science and ICT) (No.CAP22041-000) & National Research Foundation of Korea (NRF) grant funded by the Korean government (No. RS-2022-00144201) & the Nuclear Safety Research Program through the Korea Foundation of Nuclear Safety (KoFONS) using the financial resource granted by the Nuclear Safety and Security Commission (NSSC) of the Republic of Korea (No.2205013)."

**Additional Editor Comments:**

Title = not ok. It didn't reflect all the written tittle especially the "implication for rad.....". Please revise.

Abstract - Reader would like to know in first 3-4 lines the objectives, statement of problems or maybe the application that best suited this Zr-Bi-Mn-borate glass. No conclusion has been drawn in the abstract.

Abstract, line 5 - We chose nine medical linacs based on.......

Line 3 - TomoTheraphy? Or just tomotheraphy with small letter?

Line 5 - remove "Typically, 6, 10, and 15 MV energies are used during photon beam therapy using a linac for treatment"

Line 4 - 103 all in South Korea? Need rot spoecify.

Line 6-7 - Revise with full stop. Please consider active voice without the bracket.

Line 8-9 - Photon-induced radionuclides? Photon-induced activity phenomenon?

Line 11-12 - "that emits gamma rays" and exposes the worker/patiuent with unnecessary few hundred nGy/hr of gamma radiation dose? Need to elaborate the consequence of photon-induced activity phenomenon.

Line 36 -37 - Clarify your aims and objective of the work. the sentence "by assessing the activation phenomenon doest reflecting the work procedure of identifying photon-induced radionuclides and measuring gamma dose rates.

Line 77 - spectra were measured measured or analysed??

Methodology - please elaborate the HPGe detector calibration in details for both energy and efficiency calibration. It is inapproriate to use only one source of energy calibration from Co-60. If it6 is sufficient to use only Co-60 then explain why? Because of 2 gamma peak; 1.17 MeV and 1.3 MeV?

Dose rate measurement - please explain the type of survey meter? Scintillation detector NaI for gamma dose rate? Or gas-filled for alpha beta detection? Please provide the image of in-situ measurement. What its the distance of mesurem,mnet from sources?

Results and discusission - nuclide identification and dose rate need to be compared with other works. Please add a table and do more comparison.

Fig. 2 - not clear, please provide big scale and good resolution.

Linear accelerator no. - please change to no. of reference no alphabets.

Line 126 - 136 - please include "gamma" to indicate the gamma dose rate.

Fig.3 - provide big scale of figure.

Line 170 - how do you calculate the annual exposure dose? provide the Equation or appropriate dose conversion factor from any study or from Monte Carlo simulatio. From what distance? Is this annual effective dose?

Line 185 - 191 and Line 197 - 201 - redundant.Its has been elaborated in intro.

Line 210 - justify why it is needed to identify the usage history.......? Why?

Conclusion - please write a new conclusion to conclude. Don't summarise the whole work.

Reviewers' comments:

Reviewer's Responses to Questions

**Comments to the Author**

1. Is the manuscript technically sound, and do the data support the conclusions?

Reviewer #1: Yes

Reviewer #2: Yes

2. Has the statistical analysis been performed appropriately and rigorously? 

Reviewer #1: Yes

Reviewer #2: Yes

3. Have the authors made all data underlying the findings in their manuscript fully available?

Reviewer #1: Yes

Reviewer #2: Yes

4. Is the manuscript presented in an intelligible fashion and written in standard English?

Reviewer #1: Yes

Reviewer #2: Yes

5. Review Comments to the Author

Reviewer #1: The author has provided a wonderful study about the radiation dose rate at the exit window. It would be interesting if the author has data set not only at the exit window but around the head of the gantry as well. Did they perform the survey meter only at the exit window or for the sorrounding gantry as well?.

Reviewer #2: The study is relevant because the standards for handling radioactive materials in Korea are unclear.

The proposed method will be potentially useful as a general basis for establishing standards for linac radioactivity assessment and safety management

6. PLOS authors have the option to publish the peer review history of their article (what does this mean?). If published, this will include your full peer review and any attached files.

Reviewer #1: **Yes: **Anu Bhattarai

Reviewer #2: No

---

## [Author Response · Author response to Decision Letter 0]

22 Dec 2023

Thank you for your valuable feedback. We have carefully considered the comments from the reviewers and editor and made revisions to our paper accordingly. We have prepared a document titled "Response to Reviewers" and uploaded it. We greatly appreciate your input in helping us improve the quality of our paper.

---

## [Editor Report · Decision Letter 1]

24 Jan 2024

PONE-D-23-29071R1Assessment of Radio-Activation Using Spectroscopy in Medical Linear AcceleratorsPLOS ONE

Dear Dr. Choi,

Thank you for submitting your manuscript to PLOS ONE. After careful consideration, we feel that it has merit but does not fully meet PLOS ONE’s publication criteria as it currently stands. Therefore, we invite you to submit a revised version of the manuscript that addresses the points raised during the review process.

We look forward to receiving your revised manuscript.

Kind regards,

Mohamad Syazwan Mohd Sanusi

Academic Editor

PLOS ONE

Journal Requirements:

Additional Editor Comments:

Dear author,

Before we can accept the submitted article for publication in the journal, we would like to invite you to make minor revisions. Please find the following list of comments for your second revision:

Line 133: The FH 40 G-L energy detection range spans from 36 keV to 1.3 MeV. Technically, gamma emitters above 1.3 MeV, such as Sb, Mn, Br, Ni, etc., are not being measured. Therefore, the results of the measured gamma dose rates in Fig. 4 might be underestimated. Additionally, the author did not address the beta contribution. If beta is not considered in this work for occupational exposure, please highlight or make assumptions regarding why beta is considered negligible or low energy. Since the measurements were taken at 0 cm from the source, high-energy beta might contribute in FH 49 G-L, maybe? Furthermore, no other details are provided for the ESM 40 G-L (energy linearity, performance calibration, alpha-beta discriminator).

Table 3: Are the results for Canberra or Ortec? Please combine the table to present a comparison, as this work assesses qualitative rather than quantitative aspects.

TABLE 4: Instead of marking "D" in the table, provide the true detected energies. The title of the table must start with a lowercase letter.

Unit of uSv/h: Please revise. Do not use italics. Use Times New Roman. Additionally, the R^2 should be in italics.

Line 229-231: Please do not write linear equations and text in the body of the article. Use symbols 'X' and '='. Do not italicize the text. How do you consider 8 hours of exposure? Please justify, and if there is solid evidence of workers sitting close to the gantry head, provide references or any reports.

Text style: Use "Justify." Maintain the same font size. Ensure that affiliations are presented in the same size.

---

## [Author Response · Author response to Decision Letter 1]

10 Mar 2024

Dear Editor and reviewer,

I am writing in response to your recent correspondence regarding our manuscript submitted to PLOS ONE. 

We appreciate your time and the constructive feedback provided by the reviewers.

As per your request, we have meticulously addressed each point raised during the review process and have made revisions to our manuscript. 

We believe that these modifications greatly improve the quality and clarity of our work, and hope that it now meets the publication criteria of PLOS ONE.

We have uploaded the following documents as requested.

*Response to Reviewers: addresses points raised by the academic editor and reviewers.

*Revised_Highlighted_Manuscript: Manuscript that highlights the changes made to the original.

*Revised_Final_Manuscript

---

## [Editor Report · Decision Letter 2]

17 Mar 2024

Assessment of Radio-Activation Using Spectroscopy in Medical Linear Accelerators

PONE-D-23-29071R2

Dear Dr. Choi,

We’re pleased to inform you that your manuscript has been judged scientifically suitable for publication and will be formally accepted for publication once it meets all outstanding technical requirements.

Kind regards,

Mohamad Syazwan Mohd Sanusi

Academic Editor

PLOS ONE
---

## [Editor Report · Acceptance letter]

27 Mar 2024

PONE-D-23-29071R2 

PLOS ONE

Dear Dr. Choi, 

I'm pleased to inform you that your manuscript has been deemed suitable for publication in PLOS ONE. Congratulations! Your manuscript is now being handed over to our production team.

Kind regards, 

on behalf of

Dr. Mohamad Syazwan Mohd Sanusi 

Academic Editor

PLOS ONE